# High Emergence of Multidrug-Resistant Sequence Type 131 Subclade C2 among Extended-Spectrum β-Lactamase (ESBL)-Producing *Escherichia coli* Isolated from the University Hospital Bratislava, Slovakia

**DOI:** 10.3390/antibiotics12071209

**Published:** 2023-07-20

**Authors:** Ján Koreň, Michal Andrezál, Elham Ozaee, Hana Drahovská, Martin Wawruch, Adriána Liptáková, Tibor Maliar

**Affiliations:** 1Institute of Microbiology, Faculty of Medicine, Comenius University, University Hospital Bratislava, 81108 Bratislava, Slovakia; jan.koren@fmed.uniba.sk; 2Department of Molecular Biology, Faculty of Natural Sciences, Comenius University, 84215 Bratislava, Slovakia; michal.andrezal@uniba.sk (M.A.); elham.ozaee@uniba.sk (E.O.); hana.drahovska@uniba.sk (H.D.); 3Institute of Pharmacology and Clinical Pharmacology, Faculty of Medicine, Comenius University, 81108 Bratislava, Slovakia; martin.wawruch@fmed.uniba.sk; 4Department of Chemistry, Faculty of Natural Sciences, University of Ss. Cyril and Methodius in Trnava, 91701 Trnava, Slovakia; tibor.maliar@ucm.sk

**Keywords:** antimicrobial resistance, clade and subclade, ESBL-producing *Escherichia coli*, healthcare facilities, ST131

## Abstract

The expansion of sequence type 131 (ST131) extended-spectrum β-lactamase (ESBL)-producing *Escherichia coli* (*E. coli*) represents major worldwide challenges. *E. coli* strains originating from healthcare facilities (labeled No. 1 and No. 2) of the University Hospital Bratislava (UHB) were analyzed for ST131 emergence, including its (sub)lineages and clonal relatedness. Antimicrobial resistance was determined in most strains. Of a total of 354 *E. coli* strains, 263 (74.3%) belonged to ST131; of these, 177 (67.3%) were from No. 1. Generally, among 260 ST131 *E. coli*, clades A/B were confirmed in 20 (7.7%), while clade C was noted in 240 (92.3%) strains; within them, subclades were detected as follows: C0 (17; 7.1%), C1 (3; 1.2%), and C2 (220; 91.7%). Among fifteen randomly selected *E. coli* strains that were investigated for ST and clonal relatedness, seven STs were identified: eight (53.3%) ST131, two (13.3%) ST73, and one each (6.7%) of ST10, ST12, ST14, ST1193, and ST1196. From No. 1, two ST131 in the first internal clinic and one ST131 from No. 2 in the aftercare department were highly clonally related, suggesting possible epidemiological association. Antimicrobial resistance was as follows: ciprofloxacin 93.8%, ceftazidime 78.4%, meropenem 0%, fosfomycin 2.9% and nitrofurantoin 1.4%. Prevention of ESBL-producing *E. coli* dissemination, especially for ST131 clade C2, is inevitably necessary for reducing drug resistance and decreasing healthcare-associated infections.

## 1. Introduction

The global emergence of multidrug resistance (MDR) to extended-spectrum β-lactamase (ESBL)-producing strains represents an ongoing rising threat of healthcare-associated infections contributing to the antimicrobial resistance crisis. The potential driver of this might be ineffective early empirical therapy resulting in the overuse of reserve drugs required for difficult-to-treat pathogens [1,2]. In addition, the elevation in MDR strains is correspondingly related to the overall increase in the administration of antimicrobial agents, which affects the composition of the microbiota in humans [3]. *Escherichia coli* (*E. coli*) strains have been recognized as some of the most prevalent ESBL-producing Gram-negative pathogens [4,5,6] from the *Enterobacterales* order. The prevalence of ESBL-producing *E. coli* strains has increased in clinical samples to more than 10% in 22 European countries in previous years. Increased colonization in the gastrointestinal tract may have contributed to this rise even in countries that until now had a low proportion of ESBL strains, such as Sweden and the Netherlands [7]. Recent molecular epidemiologic studies have confirmed the increasing tendency for both hospital-acquired and community-acquired extraintestinal infections, especially urinary tract and bloodstream infections caused by ESBL-producing *E. coli* with sequence type (ST) 131, which was more frequent in human samples than other STs [6,8]. In particular, urinary tract infections caused by ST131 were more often persistent, had an unfavorable treatment outcome, and were related to urinary catheterization, diabetes mellitus, immunodeficiency, and higher age [1]. On the other hand, *E. coli* strains were the leading cause of bloodstream infections, and ESBL production was recorded up to 17%, while ST131 was reported to be one of the most frequent clones implicated in colonization, hospital transmission, morbidity, and mortality [9]. Accordingly, we can emphasize that these mostly extraintestinal pathogenic *E. coli* (ExPEC, from extraintestinal sites) strains are concerning for public health due to being important etiological agents, and they are most often associated with the pandemic clonal group denoted as ST131 that has resistance to fluoroquinolones and third-generation cephalosporins [10,11]. Nowadays, ST131 is associated with many MDR infections in developed and developing countries [1]. The occurrence of ESBL-producing *E. coli* strains was recorded in one-third (34.2%) of clinical samples in recent years in Slovakia [12], but research regarding the emergence of high-risk clones such as, e.g., ST131 or others, has not yet been published from our country, as they were confirmed in the former common state of the Czech Republic [13].

Expansion of the ST131 clone is commonly related to plasmid-disseminated *bla*_CTX-M_ genes, including genetic determinants of fluoroquinolone resistance. The H30 subclone of ST131 is defined according to its allele 30 of *fimH* (fimbrial adhesin-encoding gene type 1), and this genetic form has become the most prevalent [14,15,16]. Other *fimH* allele variants of the ST131 clone correspond to distinct major lineages or clades: A/H41, B/H22, and C/H30. The main generator of antimicrobial resistance is *E. coli* from the predominant clade C, mostly comprising subclades C1 and C2 (formerly H30R and H30Rx) [10,17,18]. The latter, subclade C2, is mainly associated with *bla*_CTX-M-15_ and sublineage C1, which can be related to *bla*_CTX-M-27_ (cluster C1-M27) or not (cluster non-C1-M27). The variant C0 is regarded as the progenitor of subclades C1 and C2 [10,19]. 

The objective of this study was to investigate the prevalence of ST131 among ESBL-producing *Escherichia coli* strains in two healthcare facilities within the University Hospital Bratislava (UHB). Further, the representation of ST131 lineages or clades A, B, or C was determined, and within the last (clade C), subclades C0, C1, and C2 were detected using the multiplex PCR method. Selected ESBL- and/or AmpC β-lactamase-producing *E. coli* strains, including nonproducers from both healthcare facilities, were also explored for genetic relatedness by the next-generation sequencing method. In addition, susceptibility to antimicrobial agents was ascertained in the examined strains. The resistance profile was evaluated and used to statistically compare the resistance rate associated with ST131 and non-ST131 in mostly ESBL-positive *E. coli* strains. Finally, binary logistic regression was used to examine patient- and *E. coli*-related characteristics associated with increased antimicrobial resistance.

## 2. Results

### 2.1. Patients’ Characteristics and Isolation of ESBL-Producing E. coli Strains

In total, 354 ESBL- and/or AmpC β-lactamase-producing *E. coli* strains were isolated from patients residing in two healthcare facilities within the UHB. The studied patient cohort comprised 235 (66.4%) females, and their mean age was 80.4 ± 11.9 (range from 9 to 98) years. According to the Mann–Whitney U test, the difference when comparing both genders, including age, was statistically significant (*p* ˂ 0.001). Overall, out of all patients, an age of 65 or over was predominant in 310 (87.6%) individuals, as shown in Table 1. Healthcare facility No. 1 had a larger number of ESBL-producing *E. coli* strains, with 243 (68.6%) originating here; most of them, that is, 190 (53.7%), were from the first internal clinic. In healthcare setting No. 2, 111 (31.4%) examined strains were isolated; a majority of these, 76 (21.5%), were mainly from the geriatric clinic. Regarding the type of specimen, the urine sample represented the largest number at 227 (64.1%). Specimens collected from the skin and soft tissue were less frequent, with 50 (14.1%) being found, and they were associated with a wound, ulcus cruris, abscess, fistula, skin lesion or defect, or pattern (lungs and spleen swab) obtained via the postmortem of dead patients from the Institute of Pathological Anatomy. There were 29 (8.2%) hemocultures obtained. From the genital tract, two (0.6) samples, one a vaginal swab and one ejaculate, were collected. In the cases of catheter-related infections, the four (1.1%) tips of the catheters mainly originated from the surgical site of infection. Six (1.7%) sputum samples were collected. Finally, 22 (6.2%) throat or nasal swabs and 14 (4%) stools or rectal swabs were sampled. According to defined criteria conditions, including a significant microbiological finding, our set of evaluated clinical strains was associated with 287 (81.1%) cases of infection and 67 (18.9%) cases of colonization.

### 2.2. Occurrence of ST131 and non-ST131 ESBL-Producing E. coli Strains 

A total of 354 multidrug-resistant and mostly ESBL-producing *E. coli* strains were investigated for the presence of ST131. Suspected pathogens were subjected to ST131 clade multiplex PCR with *fimH* detection [10]. Overall, 263 (74.3%) ST131 and 91 (25.7%) non-ST131 mostly ESBL-producing *E. coli* strains were obtained. ST representation in males was as follows: 86 (32.7%) ST131 strains and 33 (36.3%) non-ST131 strains. Conversely, females had a higher number of ST131 strains at 177 (67.3%) than non-ST131 strains at 58 (63.7), without statistical significance, as shown in Table 1. In addition, an increased prevalence of 236 (89.7%) ST131 strains in comparison with 74 (81.3%) non-ST131 strains was confirmed in patients aged 65 years and older with statistical significance (*p* = 0.036). Considering healthcare facility No. 1, the proportion of STs was as follows: 177 (67.3%) ST131 strains and 66 (72.5%) non-ST131 strains, but within some clinics, e.g., the first internal clinic and neurological clinic, a slightly higher rate of ST131 versus (vs.) non-ST131 was recorded, as shown in Table 1. On the other hand, in healthcare facility No. 2, a higher number of ST131 strains compared to non-ST131 were detected, being 86 (32.7%) vs. 25 (27.5%), respectively, and also within individual workplaces, such as the geriatric clinic, long-term care department, and aftercare department, a raised rate of ST131 compared to non-ST131 was found, as presented in Table 1. However, it was not statistically significant. Regarding the specimens, ST131 was more common than non-ST131, e.g., in urine samples (172; 65.4% vs. 55; 60.4%), blood cultures (25; 9.5% vs. 4; 4.4%), and throat or nasal swabs (17; 6.5% vs. 5; 5.5%).

### 2.3. Distribution of Clades and Subclades of ST131 ESBL-Producing E. coli Strains

Out of 263 ST131 ESBL-positive *E. coli* strains, 260 were used to determine lineages or clades A and B (A/B) and C, including subclades (C0, C1, and C2). Three ST131 strains remained undetermined. Clades A/B were recorded 20 times (7.7%), while clade C was determined for 240 (92.3%) ST131 *E. coli* strains. Within clade C, 17 (7.1%) cases of subclade C0 were identified, and there were 3 (1.2%) strains of C1 and 220 (91.7%) pathogens belonging to subclade C2. Regarding both genders, clades A/B had the same distribution (10; 50%), but the following subclades had a higher representation in females: subclade C0 (13; 76.5%) and subclade C2 (152; 69.1%), while a higher rate of subclade C1 was noted in males (2; 66.7%), as indicated in Table 2. The prevalence of identified clades and subclades in patients aged 65 and over was higher—A/B (16; 80%), C0 (16; 94.1%), C1 (3; 100%), and C2 (200; 90.9%)—than in younger patients. From the viewpoint of healthcare facility No. 1, and mainly from the first internal clinic, there were 12 (60%) detected strains of clades A/B and the same number, 12 (70.6%), of subclade C0. However, most of the 116 (52.7%) ST131 ESBL-producing *E. coli* strains were from subclade C2 at the first internal clinic. In healthcare facility No. 2, particularly at the geriatric clinic, four (20%) cases of clades A/B, three (17.6%) cases of subclade C0, and two (66.7%) C1 occurrences were determined. However, the highest number of *E. coli* strains, 49 (22.3%), were identified to be subclade C2 within the same geriatric clinic; further data are presented in Table 2. Based on the urine samples, 15 (75%) ST131 ESBL-positive *E. coli* strains were determined as clades A/B, and 11 (64.7%) agents were identified as subclade C0; however, most cases, 142 (64.5%), were found to be subclade C2. 

### 2.4. Genetic Relatedness of ESBL-Producing Sequenced E. coli Strains

Fifteen *E. coli* genomes were sequenced using next-generation sequencing (NGS) Illumina technology. High-quality contigs were obtained for all of the sequenced strains, with an average of 116 contigs and an average coverage of 77 times per sequenced genome. The length of genomes ranged from 3.7 Mbp to 5.3 Mbp.

The clonality of the strains was confirmed using MLST, which was used to determine the occurrence of seven sequence types (STs). Most samples (53.3%) belonged to ST131, followed by ST73 (13.3%). The remaining samples were assigned to ST10, ST12, ST14, ST1193, and ST1196 (each 6.7%).

The mutual relatedness of strains assigned to the same ST was later evaluated using core genome MLST (cgMLST). Strains belonging to ST131 showed high variability, with up to 659 different alleles between them (Figure 1). According to this analysis, the ST131 strains created seven unique profiles. Four strains (KMB-999, KMB-981, KMB-979, and KMB-984) differed only in 0–13 alleles and were clustered in one group. The rest of the ST131 strains were clonally unrelated and quite distinct, except for two ST131 strains (KMB-1000 and KMB-1002) from the urological outpatient clinic with high clonal relatedness. Seven remaining STs formed distant and unrelated forms, including the two ST73 strains, which mutually differed in 227 alleles (Figure 1).

### 2.5. Occurrence of STs in Healthcare Facilities

Fifteen randomly selected *E. coli* strains originating from two healthcare facilities following sequencing were assigned to STs. A total of 12 strains belonged to healthcare facility No. 1, of which the most ST131 strains (4) were from the first internal clinic, including 1 ST10 and 1 ST73 strain. From healthcare facility No. 2, three *E. coli* strains belonged to different STs (ST73, ST131, and ST1193). They originated from various hospital departments, including a clinic, as illustrated in Figure 1.

### 2.6. Antibiotic Resistance of MDR and ESBL-Producing E. coli Strains

Out of a total set of 354 *E. coli* strains, the presence of *bla*_CTX-M_ genes was investigated; 350 (98.9%) were found to be positive for the production of CTX-M extended-spectrum β-lactamase, without specifying the variant in more detail. Using the sequencing method, the production of AmpC β-lactamase was encountered in 11 (3.1%) *E. coli* strains and 7 (2%) possible causative agents were found to concurrently produce ESBL, including AmpC β-lactamase. Four (1.1%) *bla*_CTX-M_-negative strains only carried out AmpC production. One *E. coli* strain produced KPC (*Klebsiella pneumoniae* carbapenemase) from serine group A. The *bla*_KPC_ gene encoding this carbapenemase was confirmed using the PCR method at the National Reference Centre (NRC) for monitoring antimicrobial resistance in the Slovak Republic.

The susceptibility of MDR *E. coli* with ESBL production was evaluated, and the resistance rate to antimicrobial agents was compared between ST131 and non-ST131 *E. coli* strains. Resistance to ST131 vs. non-ST131 was determined to be statistically significant as follows: 79.1% vs. 67.5% to ampicillin/sulbactam (*p* = 0.037), 20.3% vs. 14.5% to piperacillin/tazobactam (*p* = 0.002), 99.2% vs. 96.1% to cefotaxime (*p* = 0.043), and 81.6% vs. 68.4% to ceftazidime (*p* = 0.042). Overall, resistance to ceftazidime/avibactam and meropenem was not confirmed, while resistance to ertapenem was recorded at 0.5% in ST131 vs. 3% in non-ST131 *E. coli* strains, and further resistance to gentamicin was found at 22.3% vs. 31.2%, respectively; however, these values were not statistically significant. Resistance to amikacin was detected in ST131 at 4.1% vs. non-ST131 *E. coli* strains at 5.3% (*p* = 0.038), and ciprofloxacin resistance was recorded at 96.7% vs. 84.4% (*p* < 0.001), respectively, with statistical significance; these data are listed in Table 3, including data on resistance to other antimicrobial agents.

In addition, the ability to predict patients’ characteristics was evaluated, and *E. coli* clades and subclades were analyzed based on the emergence of antimicrobial resistance. It was found that the male gender was associated with gentamicin resistance (*p* = 0.049), with an odds ratio (OR) of 2.029 (95% CI [confidence interval], 1.003–4.104) with statistical significance. Similarly, statistically significant relations were represented by a specimen collected from the upper respiratory tract (*p* < 0.001; OR 0.032; 95% CI, 0.06–0.159); collection from the medical setting of the urological outpatient clinic (*p* = 0.048; OR 0.09; 95% CI, 0.008–0.977); and the presence of ST131 subclade C2 associated with resistance to ciprofloxacin (*p* = 0.038; OR 6.238; 95% CI, 1.106–35.176). Resistance to ceftazidime was also predicted by a specimen collected from the upper respiratory tract (*p* = 0.001; OR 0.102; 95% CI, 0.026–0.394); by the type of specimen being sputum (*p* = 0.012; OR 0.047; 95% CI, 0.004–0.515); and also by ST131 subclade C2 (*p* = 0.001; OR 8.41; 95% CI, 2.534–27.918) with statistical significance.

## 3. Discussion

Carbapenemase-producing *E. coli* ST131 strains have been spreading and causing diseases in recent years [20], and the treatment options are limited for these extensively drug-resistant (XDR, i.e., susceptible to one or two antimicrobial classes) pathogens. On the other hand, the dissemination of a pandemic high-risk clone designated as multidrug-resistant *E. coli* ST131 is worrying, as it could lead to an emergency in the public healthcare system due to the production of ESBL with resistance to third-generation cephalosporins and other classes of antibiotics [1,10]. The drugs of choice for these ESBL-producing *E. coli* pathogens causing severe and life-threatening infections are carbapenems, but the increasing administration of these drugs, which are considered last-resort antimicrobial agents, can result in the emergence of carbapenem resistance [4].

In studies conducted in Denmark [21] and Sweden [18], in which ESBL-producing *E. coli* was isolated from various biological samples or urine, respectively, the findings were comparable to our research results. Even though the proportion of females in these surveys was almost the same (65—66%) as in our records, the median age was 14 years lower (69 years), and the proportion of ST131 strains (38%) was lower [21] compared to our data (Table 1). Lindblom et al. (2022) [18] reported a much lower median age of 65 for inpatient females in a previous examination compared to that in our survey (Table 1), in which the overall proportion of patients in the age category of 65 years and older was seven times higher than the number of younger individuals. Our results confirm the occurrence of ST131 in older patients, while the prevalence of this clone may differ according to regions and countries. 

In recent decades, many studies have emphasized the importance of the worldwide ST131 clone responsible for the spreading of antimicrobial-resistant infections with ESBL (mainly CTX-M β-lactamase) and coresistant profiles becoming dominant, including with regard to its acquisition in hospital [9] and community [22,23] settings. Subsequently, in the Lindblom et al. (2022) study, there was also a 2-fold higher ratio of patients from the community compared to inpatients [18]. In our group, there were a small number of patients from the urology outpatient clinic and other outpatient clinics (healthcare facilities No. 1 and No. 2, respectively). Regarding community carriage, which may contribute to infections, van den Bunt et al.’s (2020) study investigated the occurrence of ST131 in the gastrointestinal tract. They found that the most common risk factors were travelling related to prolonged carriage and antimicrobial administration [24]. In comparison, colonization was not detected in our outpatients. 

A survey from the United States underlines that the ST131-H30 subclone (or clade C) is more common in hospitalized patients, in more at-risk elderly male patients in long-term care facilities (LTCF) [1]. On the contrary, compared to our results, ST131 clade C (together, C0, C1, and C2) was more represented in the female gender. In association with healthcare facility No. 2, it was more prevalent in the geriatric clinic than in the long-term care department (Table 2). In addition, in a study by Martischang et al. (2021), ST131 H30 or clade C was confirmed in a lower (68.6%) number of samples that originated from LTCF patients than in our survey. The reason for the lower occurrence of clade C could be that the prevalence of this subclone has had a downward tendency since 2015 [17]. Moreover, Flament-Simon et al. (2020) demonstrated the following data of ST131 clades and subclades of *E. coli* with ESBL production: clade A, 5.1%; subclade C1, 27.9%, and subclade C2, 65.8% [25]. According to our outcomes, clades A/B were detected with similarly low incidence, and the C1 subclade was detected in very few values. At the same time, subclade C2 showed in our survey an 18.8% higher prevalence (Table 2) than was found in the above-mentioned joint research from Spain and France. A variety of STs have been reported from hospital environments in this region [25], including ST10 and the newly emerging global ST1193, which were also confirmed in the first internal clinic (No. 1) and the geriatric clinic (No. 2), respectively, but so far without epidemiological, i.e., clonal, relatedness (Figure 1). Recently, this rapid and ongoing expansion of ST1193 was described in U.S. cities related to young patients less than 40 years old [26].

Two ST131 *E. coli* strains isolated from the first internal clinic and from the urological outpatient clinic (No. 1), together with one isolate from the aftercare department (No. 2), were highly clonally related, with only zero to three different alleles between each them (Figure 1); these findings suggest a possible epidemiological transmission within the healthcare facilities of No. 1, and between No. 1 and No. 2. The other randomly selected STs occurred sporadically, and they showed a marked difference both in STs and in allele diversity and did not appear to be epidemiologically significant or genetically related. From the largest surveillance research on multidrug-resistant pathogens, including *E. coli*, Roberts et al. (2022) published data on epidemiologic confirmations that support clustering, which is defined as the overlap of patients in the intensive care unit (ICU) with another in the same cluster. The number of confirmed overlaps for *E. coli* strains (11; 50%) was lower than other nosocomial strains. However, this high value represents half of the assessed *E. coli* strains [27]. From the given data, it can be pointed out that MDR *E. coli* strains are of great importance in the dissemination between patients. In small numbers, this was ascertained by our investigation.

In an observational cohort survey from Vietnam, the genomic characteristics of MDR *E. coli* were evaluated [27]. The resistance genes *bla*_CTX-M_ (85%) were found to have the highest prevalence, while *bla*_KPC_ (13%) and *bla*_NDM_ (24%) occurred in lower but important numbers. Compared to our data, the same number of ST131 and ST1193 strains were also confirmed in our study, but *bla*_CTX-M_ genes were detected in the whole set (except for four strains) with a much higher prevalence. A more precise detection of *bla*_CTX-M_ genes was not performed. Based on subclades C1 and C2 (Table 2), the possible presence of *bla*_CTX-M-27_ or not, and *bla*_CTX-M-15_ [10,19,25], respectively, can be assumed. On the other hand, carbapenemase production was confirmed in only one strain by evidence of *bla*_KPC_ genes, which is a remarkable difference compared to the study by Roberts et al. (2022), in which almost a quarter of *E. coli* were confirmed to produce metallo-carbapenemase NDM (New Delhi metallo-β-lactamase) [27]. In line with our knowledge from this investigation, it could be highlighted that the emergence of carbapenemase-producing *E. coli* at our monitored medical facilities so far has been sporadic, but not insignificant. Correspondingly, it is necessary to consider warnings about the possible rise in carbapenem resistance even in *E. coli* strains [20]. In addition, ST361 *E. coli* that produces two carbapenemases encoded *bla*_KPC-3_ and *bla*_NDM-5_ was detected in a patient hospitalized in Switzerland, and this strain was resistant even to the most recent generations of antimicrobial agents (cefiderocol and aztreonam/avibactam). At the same time, it was sensitive to older agents (tigecycline, fosfomycin, and colistin) [28]. The situation is similar with colistin resistance. In our outcomes, only one ST131 *E. coli* strain was resistant to this last-line resort antibiotic (Table 3). In the findings of Roberts et al. (2022), resistance to colistin (4%) mediated by *mcr* genes (mediated by the colistin resistance gene) is reported with a slightly higher incidence [27]. It is crucial to be aware that although colistin-resistant strains are rare in our healthcare-associated facilities, if such a threat emerges worldwide, it may unexpectedly extend to our medical settings.

MacFadden et al. (2019) published results related to *E. coli* bloodstream infections, and they reported that inappropriate empiric antimicrobial therapy increases mortality [29]. In our data, the occurrence of *E. coli* blood infections with ESBL production was noted, of which, out of 29 patients, 25 cases were caused by ST131, and 21 of those cases were caused by subclade C2, which is mostly *bla*_CTX-M-15_ positive. Concerning our hemoculture outcomes, empiric treatment for susceptible *E. coli* may have been inappropriate. In the mentioned study [29], the proportion of ST131 (21%) resistance to ciprofloxacin (54%) was almost 50% lower, and to gentamicin (52%) it was two times higher compared to our results (Table 3). Accordingly, our data suggest that the administration of ciprofloxacin and gentamicin could be estimated as being higher and lower, respectively, in our medical facilities relating to *E. coli* ESBL-producing strains compared with the previous study.

Mahaya et al. (2022) determined resistance in *E. coli* ESBL-producing strains, while resistance to carbapenems, including colistin and fosfomycin, was not detected [19]; overall, a minimal reduction in susceptibility to these agents was observed in our results (Table 3). On the other hand, strains were 100% resistant to cefotaxime and trimethoprim/sulfamethoxazole, as well as to other resistance values [19]. In comparison, in our findings, resistance was much lower, especially against amikacin and nitrofurantoin; ciprofloxacin was the only exception, against which resistance in our evaluation had a higher value (Table 3). These differences in the resistance rate may be closely related to the consumption of particular antimicrobial agents, which may be diverse between various regions and healthcare facilities, but they may also be due to the spreading of distinct multidrug-resistant bacterial clones [30]. 

When comparing the antimicrobial resistance of ST131 vs. non-ST131 *E. coli* ESBL-producing strains, statistically significant differences were found for ciprofloxacin (93% and 63%, respectively) in Olesen et al.’s (2013) study [21], which was consistent with our results. However, the resistance rate was slightly higher in our evaluations. Likewise, the gentamicin resistance of ST131 (16%) vs. non-ST131 (35%) [21] was also at a similar ratio in our results (Table 3). If we compare the resistance rate of ST131 strains to trimethoprim and sulfamethoxazole with non-ST131 resistance, as was performed by Olesen et al. (2013), then the resistance was higher in ST131 strains. According to this comparison, it might be inferred that ST131 could have a similar proportion of resistance to the same antimicrobial agents as other STs.

The impact of predicted risk factors related to patient characteristics and pathogen properties on antimicrobial resistance was analyzed using binary logistic regression. The male gender had the highest positive predictive value for gentamicin resistance, including ST131 subclade C2, which is associated with resistance to ciprofloxacin and ceftazidime. According to MacFadden et al.’s (2019) research comparing patient risk factors, STs, and resistance genes as predictors for antibiotic resistance in *E. coli* bloodstream infections, it was found by using a logistic regression model that ST shows strong predictive discrimination that patient epidemiologic predictors may enhance; however, the highest degree of antibiotic administration exclusion allows for the identification and presence of known resistance genes [29].

The significance of this study compared to that of others is the highlighting of the elevated emergence of ST131 in diverse samples relating to infection or colonization, as well as the predominant representation of subclade C2 instead of other sublineages, which predicted resistance to the two above-mentioned antimicrobials. Moreover, we tried to find out the possible clonal transmission of ST131 in the survey, in which we were slightly successful.

The lacking in the study is that, within performed biochemical identification of *E. coli* strains, the PCR proof of these pathogens was not carried out.

Nonetheless, further examinations are required to determine the best way to make therapy more effective in monitoring and to combat antimicrobial resistance with the knowledge about the circulation of particular STs and close genetic relatedness, including the presence of their resistance determinants [31]; moreover, more studies are needed to identify when it is possible to implement tailored preventive measures to control these expanding risk clones. 

## 4. Materials and Methods

### 4.1. Healthcare Facilities, Patients, and Specimens

Extended-spectrum β-lactamase-producing *Escherichia coli* strains originated from patients hospitalized at the University Hospital Bratislava (UHB) during the period from January 2017 to May 2019. The UHB is one of the largest Slovak hospitals, comprising five healthcare workplaces. The patients resided in healthcare facility No. 1, Hospital Old Town, and No. 2, Specialized Geriatric Hospital. According to our standard rules, from the patients hospitalized at the UHB, signed informed consent of the patient was obtained that his or her biological materials and samples can be provided for scientific purposes including to present the data relating to research, and such consent is in each patient’s medical record if he or she was treated in our medical facility. Specimens were collected from various body sites concerning infection or colonization. Infection was specified by employing the treating physician’s diagnosis, which was consistent with the clinical manifestation of disease associated with ESBL-producing *E. coli* in the particular biological sample and with the significant laboratory findings. The colonization of the patients was defined as positive cultivation when there was no evidence of infection signs at the time of the microorganisms’ isolation.

### 4.2. Isolation, Proof of Bacterial Strains, and Susceptibility Testing 

Following the primary cultivation of the samples on standard media, *Escherichia coli* strains were detected using the biochemical set ENTEROtest 24 (Erba Lachema s.r.o., Brno, Czech Republic) in our microbiology laboratory. Only one unique strain per patient was included in the study. In parallel with regular cultivation, specimens were also inoculated on COLOREX^TM^ ESBL screening chromogenic media (MkB Test a.s., Rosina, Slovak Republic). If the suspected ESBL *E. coli* strains also grew on the corresponding screening plates, they were submitted to further testing by the combination disk diffusion test with cefotaxime (30 µg) ± clavulanic acid (10 µg) and ceftazidime (30 µg) ± clavulanic acid (10 µg) for proof of ESBL production, according to the EUCAST criteria [32]. However, the Carba NP test was used to confirm carbapenemase production when testing strains showed higher breakpoint values than the 0.125 µg/mL screening cut-off for meropenem and ertapenem [32,33]. 

Susceptibility testing of most antimicrobial agents was performed using the colorimetric micromethod that is commercially available as a MIDITECH panel [34], and has been currently modified to test strains against the 18 antimicrobials listed in Table 3 (classes: penicillins and cephalosporins (0.25–32 µg/mL; protected by β-lactamase inhibitor, 4 µg/mL), carbapenems (ertapenem, 0.03–4 µg/mL; meropenem, 0.125–16 µg/mL), aminoglycosides (gentamicin and tobramycin, 0.125–16 µg/mL; amikacin, 0.5–64 µg/mL), tetracyclines (tetracycline, 0.125–16 µg/mL), glycylcyclines (tigecycline, 0.03–4 µg/mL), fluoroquinolones (ciprofloxacin, 0.03–4 µg/mL), polymyxins (colistin, 0.25–8 µg/mL), and antimetabolites (trimethoprim/sulfamethoxazole, 0.125/2–4/76 µg/mL)). Meanwhile, sensitivity determination against nitrofurantoin (100 µg) and fosfomycin (200 µg) was carried out with a disk diffusion assay.

### 4.3. DNA Isolation and NGS Library Preparation

From 15 randomly selected *E. coli* strains, nucleic acid isolation was performed using the Higher Purity^TM^ Bacterial Genomic DNA Isolation Kit (CanvaxBiotech; Valladolid, Spain). DNA concentration was measured with the Qubit ds-DNA HS Assay Kit (Invitrogen; Waltham, MA, USA). Genomic libraries for sequencing were prepared using the Nextera XT DNA Prep Kit (Illumina; San Diego, CA, USA) and purified on AMPure XP magnetic beads (Beckman Coulter Life Sciences; Brea, CA, USA). The quality of the libraries was checked by electrophoresis with the Bioanalyzer 2100 (Agilent, Santa Clara, CA, USA). Paired-end sequencing with 2 × 150 bp reads was carried out on a NextSeq system (Illumina, San Diego, CA, USA).

### 4.4. ST-131 Clade Multiplex PCR with fimH-30 Detection

Detection was carried out according to the work of Matsumura et al. (2017). Amplification was performed with 1× DreamTaq Green Buffer (ThermoFisher; Waltham, MA, USA); 0.2 mM dNTP (Promega; Madison, WI, USA); the primers listed in Table 4, each 0.5 pmol; 1U Taq DNA polymerase (ThermoFisher; Waltham, MA, USA); and 2 μL of template DNA, with a total volume of 25 μL. DNA was amplified using the Biometra TAdvanced (Analytik Jena; Jena, Germany) with a thermal protocol consisting of predenaturation at 98 °C for 2 min, which was followed by 30 cycles of 98 °C for 10 s, 57 °C for 20 s, and 72 °C for 40 s, and final polymerization at 72 °C for 3 min. PCR products were loaded on 2% agarose gel stained with GoodView Nucleic Acid Stain (SBS Genetech; Beijing, China) and visualized under UV light [10].

### 4.5. Bioinformatic Processing and Genome Analysis

Raw reads obtained from the sequencer were assembled de novo via the SPAdes assembler using standard settings (Center for Algorithmic Biotechnology; St. Petersburg State University, Saint Petersburg, Russia) [35]. Annotation of the obtained contigs was performed with the online program BV-BRC (Bacterial and Viral Bioinformatics Resource Center) and RAST (Rapid Annotation using Subsystem Technology) software. BV-BRC was also used to detect antibiotic resistance genes by implementing the NDARO (National Database of Antibiotic Resistant Organisms) database. The criterion of 100% identity and 100% coverage with the reference gene was used as the threshold for gene presence. The genomes were further analyzed using the CGE database (Center for Genomic Epidemiology; Kongens, Lyngby, Denmark) and the EnteroBase database (Warwick University). GrapeTree was used to visualize strain clusters based on the *Escherichia coli* cgMLST analysis [36]. 

### 4.6. Statistical Analysis

Considering the data related to the study’s results, Pearson’s chi-square (χ2) and Fisher’s exact test were used alongside the statistical software IBM SPSS for Windows, version 29 (IBM SPSS Inc., Armonk, NY, USA). For the assessment of both tests, a *p*-value < 0.05 was statistically significant. The relevant patient- and *E. coli* strain-related factors associated with increased antimicrobial resistance were analyzed using binary logistic regression. The odds ratio and 95% confidence interval were used to quantify the estimated risk of these characteristics.

## 5. Conclusions

Through investigation of selected ESBL-producing *E. coli* strains from two healthcare facilities of the UHB, a three-fold higher prevalence of ST131 with a statistically significant difference was found, mainly in patients aged 65 and over, and particularly in females and from urine samples. Concerning the predominant ST131 clone, almost all types of investigated lineages and sublineages occurred the most in the first internal clinic of healthcare facility No. 1. By detecting genetic relatedness even with a small sample size of randomly selected *E. coli* strains, a possible horizontal transmission within healthcare facility No. 1 and a sporadic case between healthcare facilities No. 1 and No. 2 were identified in connection with ST131. Altogether, subclade C2 had the highest overall representation and was a significant predictor of ciprofloxacin and ceftazidime resistance; additionally, the male gender proved to be a positive predictive value for gentamicin resistance. Antimicrobial resistance was considerably higher in ST131 *E. coli* strains compared with non-ST131, with the only exception being some aminoglycosides, and indeed, these strains had a similar resistance profile to the same antimicrobial agents. Consequently, exploring these MDR worrisome and high-risk pathogenic clones is important, as well as their genetic relatedness, including recognizing genes that encode reduced susceptibility, which is crucial in the case of antimicrobial-resistant severe infections to help provide successful early therapy. 

## Figures and Tables

**Figure 1 antibiotics-12-01209-f001:**
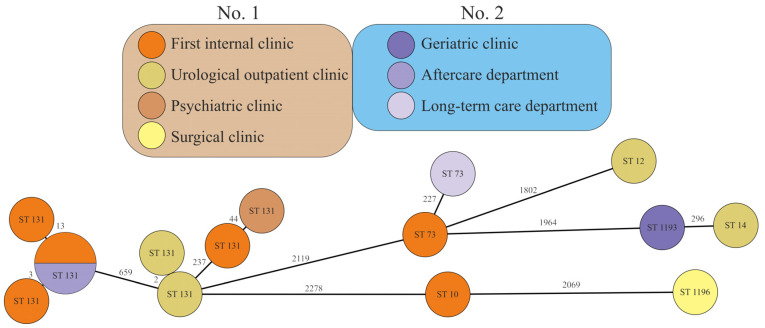
GrapeTree of 15 assayed *E. coli* strains based on cgMLST. Circle sizes are consistent with the count of strains with the same sequence types (STs) noted inside the circle. Numbers on lines between circles indicate the values of distinct alleles between groups. The colors represent the origin of the strains—clinics or departments, as well as healthcare facilities designated by healthcare setting number (**No. 1** or **No. 2**)—including clonal relatedness.

**Table 1 antibiotics-12-01209-t001:** Patients’ characteristics including the occurrence of ST131 and non-ST131 *Escherichia coli* extended-spectrum β-lactamase-producing strains in patients and healthcare settings.

		Total of (354)		ST131 (263)		Non-ST131 (91)	
Gender and Age		*n*	%	*n*	%	*n*	%
Male		119	33.6	86	32.7	33	36.3
Mean years (range) 75.0 ± 14.4 (25–96)							
Median years 79							
Female		235	66.4	177	67.3	58	63.7
Mean years (range) **80.4** ± 11.9 (9–98) *							
Median years 83							
≥65 years		310	87.6	236	**89.7 ****	74	81.3
˂65 years		44	12.4	27	10.3	17	18.7
**Setting**							
**No. 1**	First internal clinic	190	53.7	142	54.0	48	52.7
	Surgical clinic	20	5.6	13	4.9	7	7.7
	Neurological clinic	7	2.0	6	2.3	1	1.1
	Psychiatric clinic	5	1.4	5	1.9	0	0
	Dermatovenerological clinic	8	2.3	6	2.3	2	2.2
	Urological outpatient clinic	10	2.8	5	1.9	5	5.5
	Institute of Pathological Anatomy	3	0.8	0	0	3	3.3
**No. 2**	Geriatric clinic	76	21.5	58	22.1	18	19.8
	Long-term care department	18	5.1	14	5.3	4	4.4
	Aftercare department	12	3.4	11	4.2	1	1.1
	Other outpatient offices	5	1.4	3	1.1	2	2.2

***n***—number of patients, cases, or specimens; **%**—percentage; **No. 1** and **No. 2**—designations of selected healthcare facilities that belong to the University hospital Bratislava (UHB); *—***p*** < 0.001; and **—***p*** = 0.036.

**Table 2 antibiotics-12-01209-t002:** Distribution of clades and subclades of ST131 *Escherichia coli* extended-spectrum β-lactamase-producing strains relating to patients and healthcare settings.

		Total of (260)		A/B (20)		C0 (17)		C1 (3)		C2 (220)	
Gender		*n*	%	*n*	%	*n*	%	*n*	%	*n*	%
Male		84	32.3	10	50	4	23.5	2	66.7	68	30.9
Female		176	67.7	10	50	13	76.5	1	33.3	152	69.1
**Age**											
≥65 years		235	90.4	16	80	16	94.1	3	100	200	90.9
˂65 years		25	9.6	4	20	1	5.9			20	9.1
**Setting**											
**No. 1**	First internal clinic	140	53.8	12	60	12	70.6			116	52.7
	Surgical clinic	13	5	1	5					12	5.4
	Neurological clinic	6	2.3							6	2.7
	Psychiatric clinic	5	1.9							5	2.3
	Dermatovenerological clinic	6	2.3							6	2.7
	Urological outpatient clinic	5	1.9	2	10					3	1.4
	Institute of Pathological Anatomy										
**No. 2**	Geriatric clinic	58	22.3	4	20	3	17.6	2	66.7	49	22.3
	Long-term care department	14	5.4			2	11.8			12	5.5
	Aftercare department	10	3.8	1	5			1	33.3	8	3.6
	Other outpatient offices	3	1.2							3	1.4

**A/B**—clades A and B; **C0**, **C1**, and **C2**— subclades; ***n***—number of patients, cases, or specimens; **%**—percentage; and **No. 1** and **No. 2**—designations of selected healthcare facilities that belong to the University Hospital Bratislava (UHB).

**Table 3 antibiotics-12-01209-t003:** Antimicrobial resistance of total, ST131, and non-ST131 *Escherichia coli* ESBL-producing strains.

Antibiotics	Total			ST131 Strains			Non-ST131 Strains			*p **
	*n*	R	(%)	*n*	R	(%)	*n*	R	(%)	
Ampicillin/sulbactam	321	245	76.3	244	193	**79.1**	77	52	67.5	0.037
Piperacillin/tazobactam	317	60	18.9	241	49	**20.3**	76	11	14.5	0.002
Cefuroxime	320	317	99.1	243	242	99.6	77	75	97.4	
Cefotaxime	319	314	98.4	243	241	**99.2**	76	73	96.1	0.043
Ceftazidime	320	251	78.4	244	199	**81.6**	76	52	68.4	0.042
Ceftazidime/avibactam	347	0	0	262	0	0	85	0	0	
Cefoperazone/sulbactam	280	18	6.4	214	15	7	66	3	4.5	
Cefepime	281	234	83.3	214	182	85	67	52	77.6	
Ertapenem	280	3	1.1	214	1	0.5	66	2	3	
Meropenem	319	0	0	243	0	0	76	0	0	
Gentamicin	319	78	24.5	242	54	22.3	77	24	31.2	
Tobramycin	280	154	55	214	121	56.5	66	33	50	
Amikacin	320	14	4.4	244	10	4.1	76	4	**5.3**	0.038
Tetracycline	299	190	63.5	226	145	64.2	73	45	61.6	
Tigecycline	280	3	1.1	214	3	1.4	66	0	0	
Ciprofloxacin	321	301	93.8	244	236	**96.7**	77	65	84.4	<0.001
Colistin	276	1	0.4	211	1	0.5	65	0	0	
Trimethoprim/sulfamethoxazole	319	241	75.5	242	186	76.9	77	55	71.4	
Nitrofurantoin	72	1	1.4	51	1	2	21	0	0	
Fosfomycin	69	2	2.9	49	1	2	20	1	5	

***n***—number of strains; **R**—*n* of resistant strains; **%**—percentage; and *****—***p*** < 0.05 values were statistically significant in comparisons between ST131 and non-ST131 resistance rate (%), which is indicated by boldface text.

**Table 4 antibiotics-12-01209-t004:** Primers for ST131 clade multiplex PCR with *fimH-30* following Matsumura et al. (2017) [10].

Primer	Sequence	Length of Product
ST-131_F	AGCAACGATATTTGCCCATT	580 bp
ST-131_R	GGCGATAACAGTACGCCATT	
Clade C1_F	TGAATCAAAGGTCCGAGCTG	232 bp
Clade C1_R	TATGGCTGGCAGATGCTTTA	
Clade C2_F	ACGGATTCAGGTAGACGATT	164 bp
Clade C2_R	CCTCACCAAAGTTGCGATTAC	
FimH-30_F	CCGCCAATGGTACCGCTATT	354 bp
FimH-30_R	CAGCTTTAATCGCCACCCCA	

## Data Availability

Not applicable.

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
