# Peer review of "High Emergence of Multidrug-Resistant Sequence Type 131 Subclade C2 among Extended-Spectrum β-Lactamase (ESBL)-Producing Escherichia coli Isolated from the University Hospital Bratislava, Slovakia"

_antibiotics, 2023, doi:10.3390/antibiotics12071209_

Round 1
Reviewer 1 Report
In the manuscript the authors present the result of an epidemiological survey on ESBL producers E.coli isolates from inpatients of a large hospital. For that purpose, they report the prevalence of a particular clone of ESBL E.coli, so called ST131.
The survey is interesting, however several points of criticism are present.
Introduction is too short and technical, it should be integrated. Please introduce the topic more in detail . For instance, line 36-37: MDR as a general problem, not the definition. Why ST 131 should be considered as a problem? Please explain
Line 43: are ESBL E. coli mostly extraintestinal? Please explain and add reference (s).
LINE 89-91: This is the section Results, please remove this sentence (related to .. until resistance)
Line 139 and 163: 15 isolates were selected, what kind of randomisation? Why 12 from facility no 1 and just 3 from facility no.2? please explain
Discussion: The sequence of sentences should be better organised. Eg: lines 222- 225 before lines 216-221.
Other points:
Line 60 , 62 and 391 ( and in the text) : please don’t use “we”, put in third person
Line 60: delete mostly
Line 216 and 237 : please correct the sentence: in a study conducted in Denmark and Netherlans, respectively , and rephrase . As this structure is repeated in the manuscript, I suggest this revision here and elsewhere.
Line 249: compared with our results
moderate revision
Author Response
please see the attached response to your comments

Reviewer 2 Report
This study focused on evaluating the emergence of Multidrug-resistant Sequence Type 131 Subclade C2 Among Extended-Spectrum β-Lactamase (ESBL)-Producing Escherichia coli Strains Isolated from Pa-tients of University Hospital Bratislava. A total of 354 E.coli isolates were investigated, showing a considerable effort from the authors. In general, this is an interesting subject and I think some points needed to be improved before considering for publication:
- Revise the title, the detail location (Bratislava, Slovakia) should be added.
- Revise the abstract, some abbreviations should be added, the number of ST131 E.coli is not identical (line 21,22), line 25 "onetime" is not correct.
- Introduction is quite good, however lacking of paragraph update the research context in the country. The author should consider using the correct name of each healthcare unit/facilities/department.
- Results is relatively well presented. The discussion is interesting and insighful however the discussion should be simplified for a more concise scientific writing. For example, avoid using "Danish study", "Swedish study" or "Swiss study", it should be better if the authors could mention the name of first authors of these papers in an appropriate way.
- Revise the conclusion to summarize the remarks obtained results and possible perspectives.
- There are still some minor typo mistakes as well as grammatical mistakes that need to be improved.
There are still some minor typo mistakes as well as grammatical mistakes. The manuscript needed a moderate revision in English.
Author Response

(The authors gave the same response as above.)

Reviewer 3 Report
The authors declared, “High Emergence of Multidrug-resistant Sequence Type 131Subclade C2 Among Extended-Spectrum β-Lactamase (ESBL)-Producing Escherichia coli Strains Isolated from Patients Hospitalized in the University Healthcare Facilities”. Despite the importance of the study, the article lacks a good presentation. It has many grammar and language mistakes.
The order of event writing should be the same either in the abstract, introduction, material, ……….so on.
A title should be modified
The following major points must be taken into consideration:
Abstract:
-Line 39 challenge changed to challenges
-Arrange the keywords in alphabetic order and improve
Introduction:
-The introduction needs to be more informative. The introduction should be improved(illustrating the aim of the work):
· Line 38 represents changed to represent
· The objective of this study needs to be rewritten
Methods
- Authors should add a concentration of tested antibiotics in the disc diffusion method
-Authors should explain the reason for consideration of MDR and XDR in this study as mentioned in the conclusion
- They should explain the reason for their selection of antimicrobial agents tested.
- Line 387 please remove of
Results
- Line 73 years changed to the years
- Line 76 majority changed to a majority
- Line 85 was changed to were
- Line 98 was changed to were.
-Please add classes of examined antibiotic agents
Discussion and conclusion
-Authors need to improve discussion and explore the significance of the study compared to other studies.
Minor editing of the English language required
Author Response

(The authors gave the same response as above.)

Reviewer 4 Report
The manuscript is well written and findings are novel. However, I do have some suggestions for the authors
1. The title of the manuscript is too lengthy and it needs to be concise & attractive
2. In introduction, authors needs to provide some information regarding your country.
3. In methodology, I did not find any ethical and consent form information because samples were taken from human
4. Why authors did not confirm E. coli isolates through PCR also?
5. In result section, Table 1 & 2 is complicated and should be modified so that reader understand findings by first look.
Author Response
please see attached response to your comments

Reviewer 5 Report
The manuscript submitted by Koreň et al. presents results of an evaluation of predominant Escherichia coli isolates for ST and clonal relationship. The study shows that ST131 was predominant among the evaluated isolates, including the C2 clade, which has been highlighted by several studies as a contributor to the antibiotic resistance spread. The manuscript is well written, easy to understand, and the results are well presented. Below are some suggestions that should be considered by the authors:
Abstract
Lines 24-25: Please, rewrite this sentence. Suggestion: Among the 15 E. coli isolates that we determined ST and clonal relatedness, seven STs were identified: .......
Lines 30-31: Is the antibiotic resistance among clade C2 more expressive than other ones?
Results
Line: 171: Which CTX-M variant/group?
Materials and methods
Line 379: Was there ethics committee approval for data collection? Which committee did it evaluate? Please provide the protocol number.
Line 397: The method used for confirmation was the double disk synergy test or combined disk test with clavulanic acid? Which antibiotics were used to perform the test?
Line 402: It was unclear which antibiotics were used to test antibiotic susceptibility. Please include which antibiotics were tested and their respective concentrations.
Line 405-413: All isolates were submitted to WGS?
Line 415: Change “based on the work of” to “according to”.
Table 1
Correct the age range of the female patient group.

Author Response
please see the attached point-by-point response

Round 2
Reviewer 1 Report
The authors revised the manuscript on the basis of my suggestions. No further comments
Author Response
Responses to the Reviewer 1 (Round 2)
Dear Reviewer 1,
thank you for the suggested comments on the manuscript.
Dear Sir, thank you very much for evaluating and approving our manuscript.
Yours sincerely
Adriána Liptáková et al.

Reviewer 4 Report
First of all, I appreciate the efforts you put to improve the manuscript and now manuscript is more informative. However, Still I have two questions that needs to be answered.
1. Regarding ethical concern, authors said that in hospital samples were collected and microbiology laboratory is a part of hospital. In hospital we collect samples and process fs purposes but not for publishing the data. If want to publish the data then at least take consent form.
2. Authors need to confirm isolates through PCR because confirmation through biochemical tests is not enough
Author Response
Responses to the Reviewer 4 (Round 2)
Dear Reviewer 4,
thank you for the suggested comments on the manuscript.
Request: 1. Regarding ethical concern, authors said that in hospital samples were collected and microbiology laboratory is a part of hospital. In hospital we collect samples and process fs purposes but not for publishing the data. If want to publish the data then at least take consent form.
Answer: Dear Reviewer, we respect that an ethical consent form is required to publish the data. According to our standard rules, if a patient should be hospitalized at the University Hospital Bratislava, the signed informed consent of the patient was sufficient to present the data that his biological materials and samples can be provided for scientific and research purposes, and such consent is in each patient's medical record if he was treated in our medical facility.
Methods:
R: 2. Authors need to confirm isolates through PCR because confirmation through biochemical tests is not enough.
A: Dear Reviewer, we are aware that for 100% confirmation of E. coli it is necessary to perform PCR identification. The standard methodology our Institute of Microbiology of bacteriological laboratory mainly include biochemical identification; due to the incomplete confirmation, we are still performing MALDI TOF MS confirmation, but the samples (only 15 strains) that we sequenced proved to be evidence of E. coli. In our research, we used the PCR method to prove blaCTX-M genes in 350 strains (we sequenced the rest). We are aware that in the future we will already use PCR diagnostics in the identification of investigated strains.
Yours sincerely
Adriána Liptáková et al.

Reviewer 5 Report
The authors performed all changes.
So, I suggest to accept the manuscript.
Author Response
Responses to the Reviewer 5 (Round 2)
Dear Reviewer 5,
thank you for the suggested comments on the manuscript.
Dear Sir, thank you very much for evaluating and approving our manuscript.

Round 3
Reviewer 4 Report
Along with phenotypic identification, genotypic identification of bacteria is also essential. I am afraid without PCR you cannot claim bacterial identification.
Author Response
Responses to the Reviewer 4 (Round 3)
Dear Reviewer 4,
thank you for the suggested comments on the manuscript.
Methods:
Request: Along with phenotypic identification, genotypic identification of bacteria is also essential. I am afraid without PCR you cannot claim bacterial identification.
Answer: Dear Reviewer, we still think that the biochemical identification of clinical samples according to Bergey's Manual of Systematic Bacteriology is the base of reliable identification of the bacteriological laboratory of Clinical Microbiology. Confirmation using MALDI TOF MS only with a score over 2 is a high-quality identification tool. In addition, evidence of ST131 by the multiplex PCR method, in which we confirmed clades A/B and C (and subclades, C0, C1 and C2), is only possible in relation to E. coli strains according to published findings (Matsumura et al., 2017). In our data, ST131 and fimH (fimbrial adhesin-encoding gene type 1) are unique in 74.3% of E. coli. Strains that were not confirmed using the multiplex PCR genotypic method (referred to in the manuscript as non-ST131; 25.7%), had completely identical phenotypic characteristics (profile) to ST131 E. coli strains, proved by the above methods.
